# Fighting Discrimination through Sport? Evaluating Sport-Based Workshops in Irish Schools

**Louis Moustakas** [1,*] and **Lisa Kalina** [2]

1    Institute of European Sport Development and Leisure Studies, German Sport University,
     D-50933 Cologne, Germany
2    European Network of Sport Education, 1150 Vienna, Austria; lisa@sporteducation.eu
*    Correspondence: l.moustakas@dshs-koeln.de

**Abstract:** Discrimination based on ethnicity, gender, or sexual orientation remains a pressing challenge throughout Europe, including within Ireland. Despite this, anti-discrimination education is lacking and uneven within school settings. Responding to this gap and seeking to capitalise on the perceived social potential of sport, one Irish NGO has begun delivering sport-based anti-discrimination workshops to students in primary and secondary schools nationwide. This paper presents an evaluation of these workshops, putting a specific focus on the learning outcomes generated. Data were obtained from standardised, open-ended student feedback forms and qualitatively analysed using a Framework Analysis. The results illustrate fairly consistent learning outcomes, but these outcomes generally focus on individual behaviours and attitudes. This contrasts strongly with literature on anti-discrimination education, which recognises a need to reflect on privilege and social structures while also developing clear strategies to address discrimination. To conclude, we propose recommendations and ways forward to help address both individual and structural realities within such sport-based workshops.

**Keywords:** sport; racism; discrimination; intercultural education; sport for development; school sport; Ireland

## 1. Introduction

Across Europe, tackling discrimination and related behaviours, such as bullying, harassment, or violence, remains a pressing and critical issue. Indeed, numerous surveys and studies confirm that discrimination is perceived as a serious problem in society and within specific contexts such as sports or education (e.g., [1,2]). Likewise, challenges around discriminatory behaviour have become increasingly prominent in Ireland. Over the past few decades, Irish society has further diversified [3], and this growing diversity has not always been welcome. Recent survey data showed that discrimination based on ethnicity and sexual orientation is still considered widespread by a majority of respondents [4]. Elsewhere, a 2019 audit of participating schools surveyed students' and parents' experiences and needs in the context of anti-discrimination programmes. Of those surveyed, 54% said they had experiences of racism, 74% wanted action on combating racism, 60% looked for ideas on how to progress intercultural action, and 57% wanted more knowledge and appreciation of cultural diversity in their school community [5]. Despite these issues, anti-discrimination or intercultural learning across primary and high school curricula is not mandatory and remains at the discretion of individual teachers, who, in turn, do not receive obligatory and standardised anti-discrimination training [5]. Yet, according to researchers and advocacy groups, youth education is vital in shaping and changing social norms and contributing to a society with critical and well-informed citizens [6–8].

Responding to this, several non-governmental and civil society actors have emerged to advocate against discrimination and to provide educational programming within Irish

schools [3] and elsewhere in Europe [9,10]. In parallel, there has been a growing recognition of the potential role of sport-based approaches in promoting tolerance and intercultural awareness. Due to its interactive, cooperative, and practical nature and the belief that sport is a "universal cultural" manifestation, sport has increasingly been positioned as a valuable pedagogical tool to promote inclusion and combat discrimination [11–14]. In particular, many approaches used modified or adapted sport activities to generate reflection and learning about crucial social topics, as well as to support the development of skills such as tolerance, empathy, listening, and teamwork [15–18]. Research on such approaches tends to use quantitative approaches to measure intercultural competence or changes in relationships between school peers, with positive [10,19] or mixed results [20,21].

Against this background, the Sport NGO (SNGO) has been active in delivering sport-based anti-discrimination workshops for students in primary and secondary schools across Ireland. These workshops mix discussion and play-based activities to introduce primary and high school children to core concepts and to raise awareness of how to combat discrimination. The present paper, which emerged from an applied external evaluation conducted on behalf of the SNGO, specifically focuses on the learning outcomes generated by the workshops and how these affect participant knowledge or attitudes towards discrimination. In other words, the following is animated by the following question: what learning outcomes are achieved through the anti-discrimination workshops, and how do these influence participant knowledge or attitudes towards discrimination? Thus, the following work complements existing quantitative work with further, in-depth exploration of participants' learnings, attitudes, and values. In particular, this paper contributes qualitative insights from the direct target group of the workshops, thus supplementing existing work that focuses on quantitative evaluations of participants (e.g., [10,20,21]) or highlights the perspectives of educators implementing these approaches (e.g., [22,23]).

Moving forward, this paper progresses in five steps. First, additional context on current practices in anti-discrimination education, both in general and in sport, is presented to highlight emerging good practices in the area. Second, a short overview of the structure of SNGO's workshops is outlined. Third, the methodology behind the evaluation is presented in detail. Fourth, the quantitative and qualitative results are presented. Finally, the results are holistically discussed, and concrete recommendations for future improvement and development are made.

## 2. Current Practices in Anti-Discrimination Education

Various pedagogical approaches have been developed in school and sport settings to counteract discrimination and to reduce inequities in education outcomes. Though anti-discrimination education has had a long history dating back to at least the 1950s, various public incidents of discrimination and the rise of social movements such as Black Lives Matter have spurred further development in this field over the last decade [24]. Likewise, the recognition of the role of sport in social development provided by the Agenda 2030 for Sustainable Development [25] has probably also spurred the development of sport-based approaches.

Anti-discrimination educational approaches often come packaged under different names, including anti-racism, anti-bias, diversity, or intercultural education. As highlighted above, these approaches combine modified play-based activities with discussion and reflection to develop awareness and skills to tackle discrimination. Thus, these activities often occur in a mix of settings, including physical and classroom activities. Regardless of the exact terminology, research suggests there is often a disconnect between recommended practices for counteracting bias in child or youth education and the implementation of such approaches in everyday classrooms [8,24]. Likewise, evaluations measuring the long-term effects of anti-discrimination education in sport note deficits in their sustainability [26,27]. With these facts in mind, the narrative literature review below explores the practices and outcomes of various (sport) pedagogical approaches. From this, we can thus extrapolate

related good practices that can inform our later analysis and discussion of the SNGO's anti-discrimination workshops.

For one, the concept of intercultural education has become increasingly prevalent in recent years in educational settings across Europe. Broadly speaking, intercultural education can be understood as "acquiring increased awareness of subjective cultural context (world view), including one's own, and developing a greater ability to interact sensitively and competently across cultural contexts as both an immediate and long-term effect of exchange" [28] (p. 2). Essential in this is understanding one's unique socio-cultural context and the associated biases that may bring, as well as the ability to act and interact in intercultural settings. In short, intercultural education fosters a multidimensional skill set that can be promoted by culturally responsive pedagogy [29]. To achieve this holistic view of intercultural education, both the school and sport settings are seen as valuable spaces where children can come together and interact with differences and otherness. In particular, researchers and practitioners typically identify three common pedagogical characteristics as essential within sport-based approaches: mixed-group, cooperative activities; discussion and (self) reflection; and strong educator–participant relationships (see, e.g., [10,22,30–32]).

Generally speaking, anti-bias or antiracist educational work share many of the traits of intercultural education while also focusing on developing clear, concrete strategies for action [24,33]. For instance, the ideas of critical self-reflection and celebrating diversity inherent to intercultural education are often present in antiracist or anti-bias educational concepts [33,34]. As Lynch and colleagues [24] note, antiracist education should make systemic oppression visible and provoke critical self-reflection. There is no universal definition for antiracist or non-racist educational work, but generally, such pedagogical approaches represent a "politicised pedagogical approach, concerned with confronting systemic and structural oppression" [24] (p. 7). In other words, antiracist education emphasises the relationship between individual and structural discrimination [35]. The aim is to educate (young) people on the history of racist practices and to develop possibilities for action towards equality. The latter point is crucial. Developing and taking action is a core feature of antiracist education. Therefore, the goal is not only to reflect on one's position or behaviours but also to help develop strategies to tackle structural inequalities and to take concrete action [24,34,36]. Indeed, scholars from various fields argue that failing to take action, dismissing the role of privilege in discrimination, or ignoring the realities of racism through a "colourblind" approach merely allow the problem to persist [23,34,37].

Finally, in recent years, the holistic concept of a "pedagogy of diversity" has increasingly gained recognition in applied settings. A pedagogy of diversity is based on human rights, including the recognition that gender and sexual identity are not freely chosen and, in terms of gender, involve more than the division into women and men [7]. In a transparent process, the guiding principle of social diversity must be fixed in curricula and integrated into pedagogical concepts and school materials. For this type of pedagogy, teachers need diversity competence, as well as anti-discrimination strategies [7]. They need to be able to recognise prejudices and stereotypes and to reflect on them with their students. While a pedagogy of diversity fulfils the promise of the equality of every human being, it enables children and young people to be capable of acting in a self-determined way in an increasingly complex society by taking up lived diversity as a learning impulse, not denying conflicts but moderating them and making them pedagogically productive [35].

Bringing together the literature highlighted in this short narrative review, several good practices and pitfalls can be identified in the broader field of (sport-based) anti-discrimination education. To be explicit, this is not a systematic overview but rather a condensed summary of the literature highlighted, which can, in turn, serve to guide later discussion related to the learning outcomes and knowledge gained by workshop participants. As alluded to above, good practices include components such as mixed-group activities, the promotion of critical reflection around systemic factors and personal privileges, the development of clear strategies or actions, and positive relationships with educators. In contrast, pitfalls include a lack of regular engagement with the topic, repro-

duction of existing stereotypes, and inadequate follow-up. An overview of pitfalls and good practices is presented in Table 1.

**Table 1.** Overview of good practices and pitfalls in anti-discrimination education. Adapted from [8,10,22–24,28,29,31–33,35,36,38].

| Good Practice | Pitfalls |
|---|---|
| Long-term funding and integration into the formal school curriculum<br>Involvement of different public and private stakeholders<br>Workshops for teachers and staff members, as well as students<br>Inclusion of educators from diverse local communities<br>Encouragement of prominent people to share their experiences (e.g., athletes and local figures).<br>Inclusion of parents to include their voices and to raise awareness<br>Mix of class-based and sport activities<br>Diversification of sport-based activities and making them fun<br>Critical (self-)reflection on systems, intersectionality, and privileges<br>Development of strategies for structural change and allyship<br>Provision of a space for Children to reflect and to develop their own methods<br>Development of complementary awareness campaigns and digital materials | Short-term funding and ad hoc integration into the curriculum<br>Lack of follow-up, monitoring, or evaluation<br>Reproduction of **stereotypes about minority groups** among teachers, parents, students, and other school stakeholders<br>Teachers denying that racism exists and not discussing it as a topic in class<br>Forcing people, especially children, to talk about personal experiences they might have with discrimination<br>Lack of dialogue between schools and parents about discrimination |

## 3. Overview of the Anti-Discrimination Workshops

The SNGO aims to use sport and play-based activities to bring groups together and to tackle discrimination in Ireland. Beyond the workshops which we describe below, the SNGO delivers a range of regular sporting activities, special events, and educational courses for Irish and immigrant youth. In short, the organisation aims to use sport to fight discrimination and to promote social inclusion in the country. As for its anti-discrimination workshops, the SNGO generally offers one-time, 75 min workshops to primary and secondary schools, as well as occasionally to other community actors, across Ireland. Workshops are typically delivered on demand, with schools approaching the SNGO based on their pre-existing relationships with the SNGO or through word-of-mouth. As such, the schools also direct which classes or groups participate in the workshop. These workshops are designed for up to 30 participants and can accommodate youth between 8 and 16 years old. Overall, these sessions are designed around interactive discussion and game-based activities. These workshops are typically delivered by a group of two to three SNGO coaches, and these coaches generally consist of a mix of genders and include at least one person from an ethnic minority group. One senior SNGO leader responsible for the anti-discrimination programme was also always part of the delivery of the workshops.

The workshops start with a presentation of the concepts of sexism, racism and homophobia. Each presentation is accompanied by discussion questions for the group (e.g., "is there anything a boy can do that a girl can't?"). Following this, the students engage in a variety of play-based learning activities. For instance, the kids play Head/Catch, where the kids must do the opposite of what they are told. So, if the facilitator says head, the kid must catch the ball, and vice versa. In turn, this game delivers messages about thinking before you speak and listening carefully. The most prominent and notable game, however, is the discrimination game. This is a handball or football game consisting of two or four teams (depending on the number of students), and each team is assigned a captain from the most competitive or prominent students in the class. Afterwards, the captains are taken aside by a facilitator, and they discuss the qualities that make a good captain. Meanwhile, the other players are explained that their goal is not to pass the ball to their captain, and this should be kept a secret. Following the game, the secret is revealed, and captains are asked about their experience with discrimination. Likewise, the other participants are asked about their feelings and perceptions. Finally, the session concludes with a game of football3. Named after its "three halves"—a pre-match discussion, a football match, and

a post-match discussion—the football3 methodology aims to encourage communication, exchange, and conflict resolution (see [39]).

Following the game-based workshop, a follow-up is conducted in class, guided by the teachers. There, students discuss further with the teachers and fill in a questionnaire relating to the workshop. This questionnaire was developed by SNGO in collaboration with schoolteachers and includes questions relating to key learnings, favourite elements, and suggestions for improvement or action. These feedback forms, which we present in more detail below, form the basis of the data in this paper.

## 4. Methodology

### 4.1. Data Collection and Participants

To support the evaluation, open-ended feedback forms were collected from teachers and pupils following each workshop. Teacher forms featured open-ended questions regarding the appropriateness and effectiveness of the workshop, as well as space for suggestions related to curriculum integration and follow-up activities. Pupil forms captured basic demographic information and an overall satisfaction score for each respondent. In addition, space was provided for students to highlight key learnings and favourite elements of the session, to reflect on how they will put learning into practice, and to provide suggestions on future activities. Both questionnaires were created jointly between the SNGO and a group of educators and were developed with a more applied orientation. A summary of questions is provided in Table 2 below. All surveys were anonymous, and the schools facilitated obtaining consent for the collection and analysis of these data. Furthermore, for the purposes of this paper, the NGO and all schools are additionally anonymised.

**Table 2.** Summary of qualitative feedback questions.

| Questions for Pupils | Questions for Teachers |
| --- | --- |
| Write down three things that you learned during the workshop.<br>List your favourite things about the session.<br>In the future, how will you put what you learned into practice?<br>What would make the workshop even better?<br>How else could SNGO help young people to understand discrimination and put a stop to it? | Did you think the content of the workshop was appropriate for your pupils?<br>In what ways do you think the workshop was effective?<br>Have you any suggestions about other content that should be included or suggestions about how the workshop could be improved?<br>Could the workshop tie in more effectively with content already on your school curriculum?<br>We hope to produce some kind of material with ideas for teachers about how they might follow up on issues raised in the workshop. Would this be useful, and have you any suggestions? |

Two waves of workshops were held—one between February and June 2022 and another between September and November 2022. In the first wave, 1841 pupils were reached through 76 sessions across 22 different schools in Ireland. From that, 941 pupil forms and 50 teacher forms were collected. In the second phase, 109 sessions were conducted in 33 schools, reaching 2901 pupils. In this second wave, 1821 student and 58 teacher forms were returned. In total, 2762 pupil forms and 108 teacher forms were collected. Amongst these pupils, ages ranged from 8 to 18 years old, though almost 70% were between the ages of 11 and 13. In terms of gender, approximately 57% identified as male and 42% as female, with about 1% listing 'other' or 'no answer' as their response.

The input and suggestions collected through the feedback forms are the basis of the following evaluation, with additional focus put on the richer and more detailed pupil forms. In addition, to ensure adequate context and understanding of the workshops, the first author attended one workshop in March 2022 and interviewed three SNGO managers and coaches between March and May 2022. Furthermore, we consulted manuals and conceptual documents related to the preparation and delivery of the workshops, including activity guides, promotional materials, and the organisation's website.

*4.2. Data Analysis*

For open-ended qualitative answers, an analysis was conducted via a Framework Analysis. Framework Analysis was developed with applied, evaluative research in mind and involves the systematic process of "sifting, charting and sorting material according to key issues and themes" [40]. Given the applied orientation of this research, as well as the highly structured nature of the feedback forms and the volume of data collected, Framework Analysis was deemed an appropriate, rigorous approach to engage with the data collected here [40,41].

Framework Analysis contains five broad steps: familiarisation, coding framework development, coding, charting, and interpretation. These steps are not purely mechanical but are part of an iterative process that involves deep analytical thinking and revision of ideas [40,41]. The details of each step are described in the subsections below.

Before doing so, it is relevant to highlight that, given the sheer volume of data obtained, not all forms were part of the Framework Analysis. Instead, for resource and pragmatic reasons, we opted to analyse 20 sessions from the first wave, and 20 sessions from the second wave were included in the Framework Analysis. At least one session per school is represented, allowing us to include schools from all geographic, demographic, and economic settings. Follow-up workshops were not included for consistency within the analysis. (Follow-up sessions are return visits by the SNGO team to a school they have visited in the previous year. As the feedback forms ask different questions and the students have been exposed to some of the material before, these were excluded for consistency. In addition, three sessions were excluded as the digital files could not be read in the MaxQDA software.) In total, 710 feedback forms from 40 sessions were included in the qualitative analysis.

### 4.2.1. Familiarisation

Before beginning the process of coding and charting the data, the first author read a segment of the forms collected to become familiar with the responses and trends present within the data collected. Due to time constraints and the sheer volume of forms collected, about 100 feedback forms were consulted during the familiarisation stage. During this step, the first author took notes about common trends, striking responses, and preliminary reflections and began building initial ideas for codes.

### 4.2.2. Coding Framework

Following familiarisation, all notes and initial data were reviewed, and a first coding framework was developed. (Though some users of Framework Analysis use the term 'thematic framework' and indexing, to avoid confusion with other qualitative approaches (e.g., Thematic Analysis), we employ terminology related to codes and coding for the framework and indexing phases.) Here, codes should be understood as descriptive labels assigned to segments of the qualitative data that allowed us to sort and reference the data systematically.

For this evaluation, two separate coding frameworks were developed—one for teachers and one for pupils—and the codes were organised according to the overarching questions posed in the feedback forms. For teachers, an initial set of about ten codes were developed relating to appropriateness, effectiveness, and suggestions for the workshops. An initial framework of about 40 codes was developed for the pupils and focused on key learnings, favourite elements, putting learning into practice, and future suggestions. In other words, codes were created for each of the main questions posed within the feedback forms and aimed to summarise the core meaning of the answers provided. Each code developed was given an indicative title, and a short description was written to elucidate the precise meaning or purpose of the code.

### 4.2.3. Coding

In this phase, data from a larger subset of feedback forms—670 student and 40 teacher forms—were coded according to the framework developed above. MaxQDA 2022 was used to implement the coding framework, to organise the data, to assign the codes, and to support further analysis. Concretely, this means that feedback forms were read in full, and responses to the different questions were labelled with the appropriate code, allowing us to build a picture of key trends and issues embedded within the data.

As Framework Analysis is an often iterative, back-and-forth process, the coding framework was progressively revised and refined throughout. Accordingly, that meant reviewing code titles and descriptions, merging similar codes, and adjusting definitions to accommodate new insights and provide additional precision. This process led to a more fine-grained and relevant coding framework. Furthermore, the first author documented analytical or critical reflections through digital memos associated with specific data items and separately compiled thoughts, impressions, and reflections related to the entire dataset in a research diary.

### 4.2.4. Charting

Charting was organised according to the category of question asked and mapped against identified coding areas. Thus, charting involved reviewing coded segments and notes, leading to the creation of topic summaries related to overarching areas such as key learnings, favourite elements, or future suggestions. Due to the volume and nature of the data, summaries were not created for individual participants but for each session or group. These summaries were integrated directly into MaxQDA and associated with each set of forms. Given the relatively limited nature of teacher responses, many of which were limited to binary answers or simple statements, charting was only performed for pupil responses and not teachers.

### 4.2.5. Interpretation

Once charting was completed, we moved on to interpretation and theme development. This involved reviewing all codes, notes, and charts to pull together the data as a whole, identifying patterns and the meanings, relationships, and importance behind them. As such, this process moves from summarising codes to developing themes that provide shared meaning to the coded data [41]. Or, put differently, themes help build "shared meaning around a central organising concept" [42].

Theme development was achieved through two connected processes. First, data were further reviewed and synthesised by reviewing charted summaries, memos, and coded data together to generate a 'whole picture'. Concretely, this was achieved by reviewing feedback forms, consulting memos, and using MaxQDA's in-built visual tools (e.g., code maps and code relations) to explore patterns and connections. Second, theme summaries were constantly developed and refined against the charts and data available. Ultimately, this process allowed us to summarise the data against the questions within the forms and to develop three themes that expose the deeper patterns and meaning within the data.

## 5. Results

Based on the qualitative analysis of the 710 forms, we can begin to provide answers related to the questions about the effect of the workshops on participant awareness and knowledge of discrimination. The answers within each form were coded based on the above-described framework, meaning that each answer relating to learnings, favourite elements, putting learning into practice, and suggestions were labelled with a code describing the response. From this process, we can identify some of the most prominent responses across these different categories. Table 3 below summarises the top three to five codes for each category.

**Table 3.** Summary of top codes for each category. The percentage indicates the proportion of responses associated with each code within its category.

| | |
|---|---|
| **Learning** | Background and meaning (18%): Students reported learning about the background of discrimination and the definition of specific terms.<br>Don't judge or assume: (11%) Students reported learning not to judge others or to make assumptions, especially based on characteristics such as ethnicity, gender, or ability.<br>Don't be discriminatory (10%): Students reported learning about not engaging in discriminatory behaviour (e.g., racism, sexism, homophobia, and ableism).<br>Equality (9%): Students reported learning about how everyone is different but equal (e.g., 'we are one race').<br>Don't exclude (9%): Students reported learning about not excluding others. |
| **Favourite elements** | Games (42%): Students overwhelmingly appreciated the different games (e.g., handball, and head and catch).<br>Football (22%): Students identified football as a regular favourite.<br>Learning (11%): The content and learning associated with the workshop were appreciated. |
| **Putting into practice** | No practice (14%): Students did not identify any way to put learning into practice.<br>Be kind (13%): Students identified practices related to being kind, nice, fair, or respectful.<br>Support and include (11%): Students identified practices related to supporting others, standing up to discrimination, and including others.<br>Don't be discriminatory (10%: Students identified not engaging in discriminatory behaviour (e.g., racism, sexism, homophobia, and ableism) as a future practice.<br>Don't judge or assume (10%): Students reported wanting to put into practice not judging other people. |
| **Suggestions** | No suggestions (38%): Students did not identify suggestions for workshops or future actions.<br>More games or workshops (37%): Students suggested providing more workshops with more games and activities.<br>Communication activities (10%): Students suggested various communication activities, such as videos, posters, and other digital materials.<br>Examples (9%): Students suggested providing examples, role plays, or demonstrations of discrimination or how it affects people. |

Beyond this more descriptive presentation of the data, however, many patterns and trends can be identified within the data, informing the development of the themes presented below. These themes illustrate how learning across the workshops was reasonably consistent, though sometimes generated unintended consequences and did not always push pupils to develop clear, actionable strategies to tackle discrimination. The following sub-sections will present these themes along with supporting quotes, examples, and references to external literature. As such, this should be understood as a more discursive presentation format and allows the results to be situated in the context of wider research, practice, and theory related to anti-discrimination education.

*5.1. What (Not) to Do*

Literature related to anti-discrimination education or related concepts regularly highlights the need for learners to develop clear actions or strategies to tackle discrimination and warn against the negative impact of inaction [23,34]. Additionally, as illustrated in Table 3 above, several pupils identify avenues for action, especially concerning their own behaviours. Many students highlight actions such as being "friendly to everyone" (AD2a W1) or "be(ing) nice" (AD10a W1), while others refer to actively combatting discrimination by standing "up to help people" (AD26a W1). Some even identified rather original or proactive future actions, such as to "donate to charity" (AD8b), "support BLM" (AD9 W1), or "protest" (AD16a W2).

These more concrete, positive actions stand in contrast to the more passive or negative (i.e., things that one should not do) actions suggested by a significant proportion of the feedback forms. Most obviously, many students did not list any ways to put their learning into practice, nor did they suggest any improvements or future actions for SNGO. However, it is worth noting how many learnings or proposed actions centred on what pupils felt

they should *not* do. For instance, this included statements such as "never judge anyone no matter what" (AD15a) or "don't make fun of people" (AD4), as well as rather generic statements such as "don't be racist" (AD30a).

Cumulatively, these (in)actions share a common trait of centring discrimination and the ability to combat it within the individual thoughts and behaviours of the young participants. Indeed, comments relating to protesting or supporting social movements are exceptions, and most statements directly refer to personal thoughts and behaviours. Though individuals have an essential role in combatting discrimination and making educational settings more inclusive, there are risks embedded within this individual framing. Instead of recognising one's own role in discrimination and ways to combat structural discrimination, pupils focus on avoiding negative behaviours, even though they may think those behaviours are, on some level, not applicable to them or even amusing. For instance, one pupil noted that "some people make bad things fun" (AD1b), while another set felt they should not "do something if it's funny" (AD12 W1). Others felt that discriminatory behaviours did not apply to them, writing that they were "not" racist within the feedback forms (AD2 W1).

Put together, this shows that for many of the kids, combatting discrimination boiled down to a sort of behavioural abstinence, whereby they are meant to avoid engaging in actions that may be perceived negatively. As we will see next, this pattern played out especially prominently concerning language and terminology.

*5.2. Watch Your Words*

One of the most prominent learning outcomes of the workshops revolved around the background, terminology, and impact of discrimination. Pupils reported learning about the definition of certain concepts, such as sexism, homophobia or gender, for the first time, while others even talked about learning "how it feels" to be discriminated against (AD14). Such topics are often ignored within school settings [33] and, given the evident gaps around anti-discrimination education within the Irish school system [5], such initial conceptual and definitional work is likely crucial to develop awareness and activities around anti-discrimination.

Within this background information, students also reported learning about correct terminologies, such as using the term "extra-abled" instead of disabled. These comments hint at a larger pattern within the dataset, whereby pupils not only see it as essential to avoid certain behaviours but that their choice of words or terms is crucial to combat discrimination or not be seen as discriminatory. For instance, this comes across through statements such as "be careful what you say" (AD12), "think more before I speak" (AD30), or "never say something I don't mean" (AD1b). There is little doubt that words and terminology matter and, ideally, "non-discriminatory language that can be applied in multicultural contexts should be used at all times" [43]. Nonetheless, this intense semantic focus carries potential drawbacks. Not only might this hinder the development of concrete actions at the institutional or systemic levels, but it also risks overemphasising the importance of vocabulary, thus communicating the idea that discrimination can be defeated as long as our wording is correct.

Again, shared, non-discriminatory vocabulary is crucial. However, research focusing on the lived experiences of certain discriminated groups—such as youth with disabilities—suggests that using exact terminology is far less critical than actions aimed at understanding and inclusion [44].

*5.3. Trust and Tricks*

The 'discrimination game' is one of the workshops' most influential and provocative activities. Numerous students referred to "getting tricked" as one of their favourite parts of the workshop, and the game itself fed into many of the comments within the feedback forms. Through this activity, pupils directly accepted and engaged in discriminatory behaviour, leading many to reflect on how they can be influenced, tricked, or told to do

bad things. For instance, one student noted that "it is easy to be influenced" (AD2a W1), while another that "people can lead you to do something you don't know by making it fun" (AD19 W1). In later workshops, some students also noted in their reflections that they should trust themselves "to make decisions" (AD16a W2). In many ways, these first comments provide a promising avenue for further discussion. Inherent to these comments is a certain recognition of how power dynamics and structural forces can contribute to injustice and how one must critically reflect on external influences. In short, this feedback indicates how the discrimination game can provide a tangible example of the systemic or structural forces behind discrimination.

In contrast, for some, this game translated to learning not to trust or listen to others. Some statements are nuanced, such as to not "do something someone tells you if it feels wrong" (AD8b W1), while others, such as "not to trust strangers again" (AD12), are quite categorical. These outcomes are not necessarily counterproductive per se. Critically reflecting on the behaviours and words of others is a healthy and vital component of anti-discrimination education. However, in the context of these one-time workshops, such statements create an inherent tension with other stated learnings or practices. Statements such as "don't talk to strangers" (AD25 W1) present a strong contrast with ideas of accepting "that everyone is different" (AD26a W1) or "give everyone a chance" (AD10a W2). This tension underlines the inherent complexity within anti-discrimination education and hints at how students likely need further discussion, reflection, and education to make sense of this reality. Indeed, without continuous, long-term engagement with anti-discrimination education, it is unclear whether or how these tensions can be resolved.

## 6. Discussion and Conclusions

The current paper aimed to provide an applied evaluation of the SNGO's sport-based anti-discrimination workshops and, concretely, to ascertain the learning outcomes, knowledge, and attitudes of participants following these workshops. Beyond answering these questions, this paper contributes valuable insights from the direct target group of the workshops, thus expanding on existing quantitative or practitioner-focused work. As highlighted above, the SNGO anti-discrimination workshops generated fairly consistent learning outcomes across the different groups. In particular, outcomes focused mainly on the background and impact of discrimination and on identifying potential positive or negative behaviours. Given the gaps within anti-discrimination education in Ireland and the continued prevalence of discrimination elsewhere, providing this initial background and awareness is a crucial first step towards more comprehensive education on the topic. In addition, it should be noted that SNGO uses a diverse cadre of facilitators and, depending on the context of a given school, often provides the first actual contact with BIPOC individuals for many of the students. It is also worthwhile to pause on the potential implications of the learning outcomes extracted from the data. Before doing so, however, we should note some of the limits associated with our data. Though a high volume of data were collected and analysed, the answers provided were of varying depths and would benefit from further qualitative engagement, for instance, in the form of interviews or observations. The feedback forms themselves, which were designed by the SNGO in concert with local educators, feature somewhat leading questions and are not directly informed by theoretical concepts. For instance, pupils are asked how the workshop can be made "even" better, and teachers are asked, "in what ways do you think the workshop was effective". The wording thus suggests that the workshops are already good and effective and provides a limited impetus for respondents to reflect and to offer critical feedback. Qualitative surveys can provide immense value and depth (see [45]), and future evaluation work should ensure that forms should provide more open, neutral, and theoretically relevant wording. Finally, given the similarity in the statements across the forms, it is difficult to isolate the impact of demographic backgrounds, social class, or school context on the outcomes of the workshops. More in-depth work would be needed to disentangle these factors.

Returning to the learning outcomes, individual behaviours and words are clearly of great importance here. Through the three themes developed from our framework, we see that pupils mainly note behaviours or terminology to avoid, or certain supportive actions they can take. Conversely, these themes also show an absence of reflection on systems or privilege. Thus, the current focus within the learning outcomes creates the risk of reducing discrimination to something that merely lives within the words and behaviours of individuals and ignores how issues of social structures, power, and privilege play a role in discrimination. One student even quite explicitly recognised this issue, suggesting that there should be more "talking about things like privilege" (AD6b W1). Other scholars and practitioners, both in anti-discrimination education and elsewhere, likewise highlight the necessity to engage in "critical whiteness" and to develop allyship strategies [23,24,37]. As Stephanie Nixon [37] noted that social structures assign unearned privileges or disadvantages according to one's membership in different social groups. Thus, to combat discrimination and to engage in allyship, one must reflect on one's position relative to those social structures, as well as on how one's multiple identities (e.g., gender, social class, ethnicity, etc.) intersect and relate to those said structures. Critical reflection, as well as the need to centre the voices and experiences of diverse groups, are consistent recommendations throughout the broader literature [24,31,33,46].

Anti-discrimination education also aims to move beyond reflection and to generate clear, actionable strategies to tackle systemic discrimination [24,33]. Even if somewhat individual-focused, most of the students identified actions they could take, be it as it relates to not excluding others or actively supporting people facing discrimination. Nonetheless, it should be noted that many students did not identify ways to put learning into practice or suggestions for improvement. This lack of clearly identified actions suggests a need to include room for strategising and planning within the sessions more consciously, and this process should actively give room for the pupils to reflect and to determine their own solutions. Of course, addressing these gaps is likely easier said than done.

For one, though the difficulty of integrating such reflection and strategising amongst children should not be minimised, there are existing activities and concepts that could be applicable here. Work conducted by Kimura and colleagues [33], for example, demonstrated how educators can productively handle complex political topics within early childhood settings. Different (sport-based) experiential learning activities can also be employed to foster discussion. The impact of privilege can be playfully demonstrated through games such as the privilege walk [47,48]. Likewise, questions within the influential discrimination game could also be re-oriented to discuss how systems or structures can promote discrimination. In addition, several other curricula exist to promote such learning, including sport-based materials designed through European projects such as EDU:PACT, Outsport [9,49], as well as more general anti-discrimination educational approaches [7,32,35], and these could be used to support further programme development.

In reality, the challenges and limitations faced by the SNGO are shared across numerous contexts and programmes. Struggling to have participants develop strategies or recognise systemic issues are hardly unique to the SNGO programme [24], and many initiatives are restricted by their short, limited-time, and optional nature within the broader curriculum [20,32]. Of course, the SNGO could expand its programme, offer additional follow-up workshops, and develop further educational materials. Realistically, though, a few workshops cannot provide more than awareness-raising. Racism, sexism, and homophobia are structural, institutional, and systemic. The aim of anti-discrimination education should be to raise awareness, to highlight the intersectionality of discrimination, to let students reflect on their privilege, and to develop strategies for change. Such education should be included as a standard across all levels of the curriculum, including as it relates to teachers, administrators, sport coaches, and other professionals. In physical education and coaching, in particular, issues of intercultural education or discrimination are not always addressed [22,23]. This leaves many sport educators unprepared to deal with the topic, creating a real risk of reproducing stereotypes and (re)establishing whiteness as an

accepted default norm [23]. To address these issues, we must continue developing pedagogical approaches that integrate systemic perspectives, work to integrate anti-discrimination education within local and national curricula, and ensure that all levels of the educational system—not just pupils—receive such education. Thus, though our research must continue to explore the implementation and outcomes of anti-discrimination education programmes, it must also seek to understand the political barriers and opportunities surrounding the integration of anti-discrimination education within sport and general education.

In short, for those of us in the research, education, and advocacy sectors, we need to not only focus on delivering or improving individual programmes like that of the SNGO, but also advocate for systemic change within the training and delivery of education more broadly.

**Author Contributions:** Conceptualisation, L.M.; methodology, L.M.; formal analysis, L.M.; investigation, L.M.; resources, L.M.; data curation, L.M.; writing—original draft preparation, L.M. and L.K.; writing—review and editing, L.M. and L.K.; visualisation, L.M.; supervision, L.M.; project administration, L.M.; funding acquisition, L.M. All authors have read and agreed to the published version of the manuscript.

**Funding:** This research was part of an evaluation financed by the Sport NGO.

**Institutional Review Board Statement:** The study was conducted according to the guidelines of the Declaration of Helsinki. As this study was performed on a consulting basis, IRB approval was not available. All participant information, as well as the name of the NGO and schools, have been anonymised.

**Informed Consent Statement:** Informed consent for the collection and use of data was obtained from all participants.

**Data Availability Statement:** Due to the minor status of the target group collected, the data are not available in a repository. Data can be made available upon reasonable request to the corresponding author.

**Conflicts of Interest:** The authors declare no conflict of interest.

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
