# Peer review of "Fighting Discrimination through Sport? Evaluating Sport-Based Workshops in Irish Schools"

_education, doi:10.3390/educsci13050516_

Round 1

Reviewer 1 Report

Abstract

12 – to students or to staff, or admi/leaders?

13 – public school? Or others? Which levels?

20 -  sport-based workshops (Antidiscrimination worskhop)?

Keywords: consider adding Ireland, school sports,

Introduction

56-59 Add reference(s)

56-57: whom are the workshops addressed to (students, staff, school leaders?). which schools (public, both publica and private, EY primary, middle, high?)

59: ‘this/the present paper’ at the place of ‘the following paper’

2.

It would be good to add a time and place framework here: shedding some light on the historical development of the inclusion of these practices and specify regional/national focus

82. in sports (PE) classroom, or in classroom in general?

81-82 In the introduction you mention the focus of the anti-discriminatory practices from Primary to high schools, but here you mention ‘early childhood education’ (ref. 8). Could you expand this point to older children and add a reference?

84-87 this sentence should be placed rather in the methodology, as here it is still about literature review and background

140 – table 1 (sources are missing)

3.

141. the SNGO should be described further, its mission, vision, role, organization, etc.

143 how are the participants selected? Please describe more about the organization of the workshop at school. How are the schools selected?

Methodology

Very well explained.

Results

It would have been good to see some figures in terms of qualitative data translated into quantifyable measurement. How many of the interviewed individuals had the same (similar) feedback?

It would be good to highlight difference among the feedback according to students with different profiles (elementary, middle, secondary ages, gender, areas, socio/economic status, etc.

471-473To support the last recommendation, it would be good in the literature to present other initiatives other than in sport, used in schools to fight discrimination.

Another aspects that I think it is missing is the assessment of the SNGO initiative leaders: are there always the same across all the workshops? Which are their (winning and losing) characteristics?

A short conclusion would add to wrap up the article. Also mentioning about the recommendations.

Reviewer 2 Report

This is an interesting article, which explores a number of issues that may be highly relevant to the readership. I have some general comments about the “pedagogical approaches”, and methods/discussion for the author(s) to consider.

1. Introduction

Ln 21 “Irish society has changed and diversified dramatically”: by reading the manuscript I get the impression you refer to the circumstances that diversity has become a more and more discussed topic, which has been there before? Isn’t it more a question of the perspective?

Ln 65/66 “Thus, the following complements the existing quantitative work with further, in-depth exploration of the learnings absorbed by participants”: I totally agree with you, but I think this aspect needs a bit more elaboration in detail (here) and connection in the methods/discussion.

2. Current Practices

Ln 126: “gender competence as well as anti-discrimination strategies”: if you refer to a “pedagogy of diversity“ (ln 121) should it than be “diversity competence”, which comprises more aspects you address?

Ln 140: How did you bring you “these strands together” (ln 133)? Have you conducted a review for table 1? For me, this remains unclear but important for the following steps. In my eyes, you should elaborate on that.

4. Materials and Methods

Ln 173 “open-ended feedback forms”: (how) are these questions related to theory/further research. I think you must explain a link here. For me, it is the same with 4.2.1 (ln 224/225): to what extent are your notes (not) connected to further research?

Ln 243 “data from a larger subset of feedback forms – 670 student and 40 teacher forms”: what is your argument for referring to this specific subset. I think there should be a reason.

6. Discussion

I think it would strengthen the manuscript if you more clearly address how your research complements to existing quantitative research (see also ln 65/66).

Reviewer 3 Report

First of all, I congratulate the authors for the subject matter and the quality of this research's content.

In the following, I will detail some areas for improvement in the different sections of this study.

Theoretical framework

It is well-developed, with arguments and scientific evidence. It is suggested to consider the possibility of deepening the theoretical framework in the importance of this topic to solve priority objectives of sustainable development established by the 2030 agenda (United Nations).

Another aspect of improvement refers to the need to describe the objectives of this study in the last paragraph before Material and Methods (line 170)

Material and Methods,

4.1 Design research

The first section should develop the design. It should explain the type of study that corresponds to this research.

4.2 Participants

This second section should specify the characteristics of the sample (n, age range, SD, % gender, people from different cultures...). The code of approval of this study by an official ethics committee should also be detailed.

4.3 Instruments and Procedures

The third section should elaborate on the instruments and the procedure followed.

The procedure followed to validate qualitative questions should be explained (e.g. expert opinion; modifications concerning the first version...).

4.4 Data Analysis

This section is adequately explained, but the explanation should be complemented by identifying the dimensions and variables to be studied per the objectives.

Likewise, more data should be provided on whether any programme (software) was used for the qualitative analysis (e.g., Atlas. Ti; nudist...).

To encourage other future studies by the scientific community (researchers from different academic societies, universities, etc.), it should be indicated whether the database obtained is in a repository so that other researchers can use it in the future.

5. Results

As far as possible, it is suggested that this section or the discussion section include a diagram or figure to facilitate understanding the different aspects studied and the main findings.

6. Discussion

The first paragraph should begin by recalling the objectives of this research (line 397).

The authors should include other issues, such as:

7. Conclusions and future perspectives

This section should specify the main conclusions of this research. It is also suggested that the authors include in this section the study's limitations and the outlook for future research.

Round 2

Reviewer 3 Report

I congratulate the authors for this last version. The changes they have introduced improve the quality of the manuscript.